# Enhanced Multicast Repair Fast Reroute Mechanism for Smart Sensors IoT and Network Infrastructure

**DOI:** 10.3390/s20123428

**Published:** 2020-06-17

**Authors:** Jozef Papan, Pavel Segec, Oleksandra Yeremenko, Ivana Bridova, Michal Hodon

**Affiliations:** 1Department of InfoCom Networks, University of Žilina, 010 26 Žilina, Slovakia; pavel.segec@fri.uniza.sk (P.S.); ivana.bridova@fri.uniza.sk (I.B.); 2Department of Infocommunication Engineering, Kharkiv National University of Radio Electronics; 61000 Kharkiv, Ukraine; oleksandra.yeremenko.ua@ieee.org; 3Department of Technical Cybernetics, University of Žilina, 010 26 Žilina, Slovakia; michal.hodon@fri.uniza.sk

**Keywords:** Internet of Things (IoT), ReRoute, Multicast Repair (M-REP)

## Abstract

The sprawling nature of Internet of Things (IoT) sensors require the comprehensive management and reliability of the entire network. Modern Internet Protocol (IP) networks demand specific qualitative and quantitative parameters that need to be met. One of these requirements is the minimal packet loss in the network. After a node or link failure within the network, the process of network convergence will begin. This process may take an unpredictable time, mostly depending on the size and the structure of the affected network segment and the routing protocol used within the network. The categories of proposed solutions for these problems are known as Fast ReRoute (FRR) mechanisms. The majority of current Fast ReRoute mechanisms use precomputation of alternative backup paths in advance. This paper presents an Enhanced Multicast Repair (EM-REP) FRR mechanism that uses multicast technology to create an alternate backup path and does not require pre-calculation. This principle creates a unique reactive behavior in the Fast ReRoute area. The enhanced M-REP FRR mechanism can find an alternative path in the event of multiple links or nodes failing at different times and places in the network. This unique behavior can be applied in the IoT sensors area, especially in network architecture that guarantees reliability of data transfer.

## 1. Introduction

The Internet of Things (IoT) model allows the connection and exchange of data between various types of smart devices. These smart devices, usually sensors, can be connected in the Wireless Sensor Network (WSN) [1,2,3,4] creating a unique sensor architecture [5,6]. With increasing numbers of sensors in the environment and the importance of measured data, the network platform must guarantee the reliability of connection.

Historically, Internet Protocol (IP) networks have been focused mainly on time-tolerant communication services, such as e-mail, file transfer and access to web content. However gradually, IP networks have evolved into converged platforms supporting several different types of services, including time-consuming and real-time applications such as voice transmission over IP, Internet of Things platform, sensors, streaming and multimedia services [7,8]. These services have higher network performance requirements, such as delay, availability, or packet loss, and are also negatively affected by unexpected link or node failures in the network. In case of network failures, network routing protocols (IGP), such as Open Shortest Path First (OSPF), respond to network failures by flooding topology updates and calculating new routes [7,8,9,10,11]. This process is also known as the network convergence process. Thus, in a period of network convergence, network routers have outdated information that can cause data loses and outages.

For this reason, IP networks must meet specific qualitative and quantitative parameters in order to ensure an acceptable quality of service, e.g., availability and short repair time after a specific network failure. To do this, network providers must deploy appropriate technologies to ensure uninterrupted customer service. One such significant technology is a group of mechanisms for rapid network recovery, also known as Fast ReRoute (FRR) [12,13,14,15]. A key principle of FRR mechanisms is that the backup path for possible failure scenarios is calculated in advance before the failure occurs. This principle presumes that the switching to a precomputed backup path is faster than waiting for the network convergence process to complete. FRR mechanisms that are designed to work on IP networks are known as Internet Protocol Fast ReRoute (IP FRR) mechanisms.

The paper presents an enhanced version of Multicast Repair (M-REP) FRR mechanism (hereinafter EM-REP) that does not require pre-calculation of alternative routes in advance. At the same time, EM-REP is capable of finding an alternative path even in the case of multiple network failures. These features make EM-REP a unique IPFRR mechanism which can be also used in IoT architecture providing reliable connection for various types of sensor networks.

The rest of this paper is organized as follows. Section 2 provides an insight into the issues of Fast ReRoute, the purpose, and terminology used. Section 3 provides state-of-the-art analysis, where existing FRR solutions and identified problems are discussed. Section 4 presents the specification of the original M-REP FRR mechanism and Section 5 proposes its enhanced version, the EM-REP FRR mechanism. Section 6 focuses on the evaluation of the EM-REP mechanism and compares its features with the predecessor, the M-REP algorithm, as well as with the other FRR solutions. Finally, Section 7 presents the conclusions and directions for future research work.

## 2. The Fast ReRoute

One of the main reasons why Fast ReRoute mechanisms have started to be used is that the process of network convergence after a network failure usually takes a longer time than that expected by the network provider.

In the context of routing, the convergence state of a network is the state in which each network router has up-to-date, complete and consistent routing information [16,17]. This means that each router is informed of the current network status (up-to-date), of each available network (complete), and each router has chosen the optimal successor/next-hop for each network according to a common criterion (consistent). The convergence process is the operation of each individual routing protocol process to achieve a state of convergence. The time required to complete the process of network convergence depends on various factors, such as the number of devices, especially network routers; topology complexity; the type of routing protocol used (link-state or distance vector); and its individual operational parameter values (like update or hello timers). The time required to finalize the network convergence process usually ranges from hundreds of milliseconds up to a few seconds. The network convergence process may consist of several subprocesses, such as failure detection, response, the distribution of updated routing information, recalculation and, finally, the installation of new routes [7,18,19,20]. The resulting time of network convergence is the sum of durations of each individual subprocesses.

First, from the router point of view, there is a need for mechanisms that perform failure detection. Failure detection is the time that the router’s operating system (OS) needs to detect a failure, i.e., a failed link (network interface) or unavailable neighbor router. Interface failure detection may take up to several milliseconds on the physical layer of a router. The detection of neighbor router unavailability depends on the type of routing protocol used. This failure detection may take from tens of milliseconds when a link state protocols are used (OSPF/IS-IS hello mechanisms), or it may take up to tens of seconds in the case of distance vector routing protocols (RIPv2). During this period, the packets affected by the failure are permanently lost due to outdated routing information and resulting incorrect routing decisions. Next, the local router response to a link failure; it generates and distributes topology/routing updates that reflect actual state. The duration depends on actual load conditions of each individual router. The distribution of information to other routers is required to inform other routers about the situation and starts within 10 ms to 100 ms for each affected next-hop router [18,19,20]. All routers that have received actual routing information must recalculate their routing tables. The recalculation of routing tables depends mainly on the size of the network and the amount of topological information. This may take a few milliseconds for link-state routing protocols that use the Dijkstra algorithm. After the recalculation is complete, routers install new routes and update their routing tables. Again, this mainly depends on the type of router and the number of prefixes that were affected by the network failure.

### 2.1. The Principle of Fast ReRoute

For the proper understanding of the FRR technology, there is a need for terminology that denotes routers with special meaning for FRR mechanisms [21,22,23,24,25]. Here, we introduce the terms using the following simplified network topology described in Figure 1. The source router (S) is a router that has detected a link or neighboring router failure and then activates a locally implemented FRR repair mechanism. In other words, router S is actively involved in FRR repair (Figure 1). This router is also called Point of Local Repair (PLR). The destination router (D) is the destination router of the original data flow. Routers N1, N2, N3 and others are specific routers that are used as an alternative next router (hereafter referred to as next-hop router) for a specific FRR alternative path. R (R1) is a router that is not actively involved in FRR repair.

However, before starting the main FRR process, an administrator must set up protected links or prefixes that are managed by the router. Subsequently, the FRR mechanism pre-calculates an alternate next-hop router to be used in the event of a protected link or prefix failure. This is called Phase Zero (preparation). FRR can then proceed further through the following phases [26,27,28]:Phase One: Detection of a link failure by the specialized FRR technology. This phase activates the FRR mechanism. In the Figure 1, the Fast ReRoute process starts after a failure of the link between routers S and E has been detected. Here, following the terminology, the router S detects a link failure.Phase Two: Temporary modification of affected routing records by the FRR mechanism. During this phase, precalculated alternative routes are being installed (the FRR mechanism is active).Phase Three: Performing background routing protocol update. Routes installed using the FRR mechanism are used to route packets until the network convergence is completed (the FRR mechanism is active).Phase Four: The routing protocol completes the necessary routing information update. As the next step, the FRR mechanism is deactivated and the routing process is taken over by the routing protocol.

Once the update of routing information is completed, the deactivation of the FRR mechanism can be accomplished in several ways. One method used is to apply a hold-down timer. This timer should be set to a minimum time necessary to complete the network convergence process. After this timer expires, the temporary routing information installed by the FRR mechanism is removed and the FRR mechanism is subsequently deactivated [26].

The main advantage of the Fast ReRoute mechanism is that it offers several times faster network transmission recovery than a traditional routing protocol may achieve (OSPF). The average repair time of actual FRR mechanisms is up to 50 ms [26,29,30,31].

### 2.2. Precomputation Approach of Fast ReRoute

A key feature common to Fast ReRoute mechanisms is that they calculate the backup path in advance and therefore offer faster network recovery [32,33]. The precalculated backup path in the FRR terminology is also referred to as a precomputed backup path [34,35]. To ensure correct network recovery, the backup path cannot pass through the failure point. Depending on the FRR mechanism, a given router may also calculate several backup paths. When calculating and installing a pre-calculated alternative route, each router decides independent of other routers. The principle of precalculated alternative routes is currently used by all FRR mechanisms. This proactive approach is an important factor in minimizing the time required for fast network recovery after failures [32].

## 3. Related Works

In the IoT area, several existing solutions dealing with rerouting have been proposed. In paper [36], a new approach of jamming attack tolerant routing using multiple paths based on zones is presented. The proposed scheme in that paper separates the system network into specific number of zones and directs the candidate forward nodes of neighbor zones. After detecting a specific attack, detour nodes in the network determine zones for rerouting and detour packets destined for victim nodes through forward nodes in the decided zones.

In work [37], the authors present a detailed review of IoT sensing applications in WSN and the difficulties and challenges that need to be overcome. Some of these challenges are fault tolerance, the effectiveness of the energy harvesting, communication interference, cost feasibility, and an appropriate integration of these elements.

At present, there are many unicast IPFRR mechanisms that differ in the way that alternative routes are calculated. Three of the most common and widely used IPFRR mechanisms are the Equal-Cost Multi-Path (ECMP) [30,31], Loop Free Alternates (LFA) [24,31,38] and its extended version, Remote LFA (RLFA) [30,39].

The LFA mechanism calculates alternative routes based on conditions that consider metrics for each next-hop router. These conditions ensure that if a packet is redirected to this alternate next-hop router (that has met the conditions), the router delivers the packet to the destination over a longer path that is still loop-free and bypasses the network failure. The Remote LFA is an improved version of the original LFA. The idea of Remote LFA is to use a tunneling mechanism from the source router S to the remote LFA router. The tunnel is used to bypass the part of the network that, in the event of an error, would route packets (not tunneled) from the affected site back to the source router S or would forward them through a failed link or router. The RLFA router may be a few hops away from the source router S.

Other mechanisms, although less common, are Multiple Routing Configurations (MRC) [11,30,40,41,42,43,44] and Not-Via Addresses [35,45]. Furthermore, there are tunneling-based mechanisms, such as Maximally Redundant Trees (MRT) [46,47], and, finally, IPFRR mechanisms based on alternative trees [22,48,49,50]. IPFRR mechanisms such as Not-Via Addresses, Multiple Routing Configurations and Maximally Redundant Trees can provide protection that is close to 100% of repair coverage [26,51]. The main challenge of these IPFRR mechanisms is the complexity of internal algorithms that calculate alternate paths.

There is also another FRR group of mechanisms that focus on the protection of multicast communication [52,53]. In general, these solutions utilize precomputed multicast disjoint trees. Examples of these mechanisms are Multicast Only Fast Re-Route (MoFRR) [54] and Bit Index Explicit Replication-Traffic Engineering (BIER-TE) [51,55].

We have been analyzing and researching FRR mechanisms for several years [21,56,57,58,59,60,61]. Based on the obtained results, we can summarize the most significant properties of the existing FRR mechanisms in Table 1.

### 3.1. Problem Formulation

In analyzing the FRR mechanisms mentioned above, several issues have been identified. We can classify them into the three basic problem areas, which are briefly introduced in the following subsections.

#### 3.1.1. Cost-Based Calculation of Alternative Route

The majority of existing FRR mechanisms, such as LFA [24,31,62], Remote LFA [14], Directed LFA [63], ECMP [23], MRC [11], and MRT [46,47], calculate an alternative backup route according to link metrics. Alternative routes are usually calculated using a Dijkstra SPF algorithm, which calculates the route path as the minimal total cost of each individual link. The main problem with this type of calculation is that a valid alternative route can only be calculated if the internal algorithm of the FRR mechanism is able to find the correct alternative route according to specific metric conditions. In other words, there are topologies or situations where one FRR mechanism can find an alternative path but another FRR mechanism is unable to do so. If link costs exceed mathematical conditions, the FRR mechanism cannot find an alternative route, even if the alternative route physically exists. The positive effect of the cost-based FRR mechanisms is that they guarantee the calculation of the most advantageous alternative route in the event of a failure. On the other hand, it should be noted that they depend on the correct cost of links in topology. Therefore, there is a need for an FRR algorithm that is able to find an alternative route without cost-based calculation of an alternative route.

#### 3.1.2. Single Failure Recovery

Mechanisms such as Remote LFA [39], Directed LFA [63], and Not-Via Address [45] are designed to be able to protect networks only in the event of a single failure. In situations where more than one failure occurs, these FRR mechanisms cannot create an alternative path and to reroute affected traffic around the failed element in the network. Therefore, packets could be lost in this situation, as the mechanism was not designed to account for more than one point of failure. This is sometimes identified as a limitation of these mentioned FRR algorithms.

#### 3.1.3. Dependency on Link-State Routing Protocols

Another important fact is that several analyzed FRR mechanisms require topological network information from a link-state routing protocol database to calculate an alternative path [26]. This feature limits the application of FRR mechanisms only to networks where a primary link-state routing protocol is deployed. Currently, most of the existing FRR mechanisms are dependent on information from link-state routing protocols.

#### 3.1.4. Packet Modification

The key part of fast network recovery technology is the fast detection of the failure and the subsequent means of its notification to the other routers which were affected by the failure (disrupted routing). In some FRR mechanisms, specific link failure information is distributed by the following techniques:Modifying special bits in the IPv4 header (MRC [11]);Encapsulating the packet with another header (Remote LFA [30], Directed LFA [63]);Based on the interface through which the packet was received (LFA [46,64]).

It should be noted that packet modification causes various compatibility problems as well as problems with exceeding the Maximum Transmission Unit (MTU) on some network links [26].

#### 3.1.5. Preparatory Calculations

The analyzed FRR mechanisms work on a principle which is based on the fast detection of link failure with a neighboring router and precalculated alternative routes (precomputing). The high complexity of these precalculations is a problem area [26].

The computational complexity of individual FRR mechanisms increases with the increasing number of routers in the network. These calculations must be repeated if there is a change in network topologies and they are typically performed on routers as specific low-priority processes when the router’s Central Processing Unit (CPU) is idle [26]. Thus, the FRR mechanism calculations take up the valuable time and system resources of the router. Based on these facts, we conclude that one of the problem areas of FRR mechanisms are preparatory calculations. All existing FRR mechanisms work on this principle.

Based on the problems thus identified and their problem areas, this document proposes the EM-REP FRR mechanism, an improved version of the M-REP algorithm. The main improvements of the EM-REP proposal focus on the protection of important unicast flows in the event of subsequent and recurring failures occurring concurrently over time. This is a unique feature, as we identified that existing IPFRR mechanisms provide the single failure protection.

Our M-REP algorithms are not dependent on any of unicast routing protocols in general, but EM-REP is enhanced in a way that provides an advantage in specific deployment scenarios, where the area design is used (for example OSPF or IS-IS). Here, we propose the modification of the Area Border Router (ABR) router behavior. This modification adds flexibility to optimize packet delivery through an “on-the-go” decapsulation process, compared to the old M-REP approach where it must be performed on the last router in the delivery chain. This is the second M-REP algorithm enhancement introduced in the paper.

## 4. M-REP FRR Mechanism

Based on the analysis and identified problem areas introduced in the previous section, a M-REP FRR mechanism has been proposed. The M-REP mechanism does not require precomputation of an alternative route, it is not dependent on any unicast routing protocol, it does not use a metric calculation of the alternative route, and, finally, it provides full repair coverage.

The M-REP FRR mechanism uses a multicast [65,66,67] routing protocol, Protocol Independent Multicast-Dense Mode (PIM-DM), as its basis. PIM-DM, at the beginning of multicast transmission, floods multicast packets to all PIM-enabled routers in a domain. We decided to use this flooding feature as the basic behavior of our M-REP algorithm. However, to fit PIM-DM to our purpose, we modified its Reverse Path Forwarding control mechanism.

In this section, the description of original M-REP mechanism is provided. In the next section, we will follow up with the presentation of its improved version, the Enhanced M-REP mechanism (EM-REP). EM-REP can create alternative route that allows recovery from the occurrence of multiple and even parallel failures.

### 4.1. Description of the Original M-REP Mechanism

To describe the M-REP mechanism, the role of routers in IPFRR is modified as follows:S router (source router) is a router that has detected a connection failure with its primary next-hop for a specific destination host. Router S begins to encapsulate the original unicast protected flow (or the protected flow, see Table 2) into packets of a specific multicast (S, G) flow. Here, the S address is the original address of the host, that sends packets. G is a specific, pre-configured multicast group address, that is used by the M-REP IPFRR to encapsulate packets of the protected flow. Router S becomes the root of the tree created by the M-REP mechanism.D router is a router that performs M-REP IPFRR multicast flow recovery back to the original unicast packets of the protected flow. Router D will further route and forward packets to the destination host as unicast. The destination host, i.e., the target for the original protected flow, must be directly connected or reachable through the D router.R router is a router with implemented IPFRR M-REP mechanism.

The M-REP mechanism is designed to protect only specific important customer data flows, delivered over an Internet Service Provider (ISP) network from the source S to the destination host D (Figure 2a, mark 1, red path). A router that detects a connection failure (link or node) becomes the source router S (Figure 2a, mark 2). At the same time as the failure was detected, routers start the process of network convergence. In this moment, the source router S begins to encapsulate the original unicast packets of the protected flow into a specific M-REP multicast flow identified by (S, G) pair. These multicast packets are sent directly out (utilizing the flooding process of Protocol Independent Multicast-Dense Mode protocol) on all active PIM-DM enabled interfaces of router S (Figure 2b, mark 3, dashed green arrows). This starts the process of creating a multicast distribution tree by the PIM-DM. The result of this flooding represents an alternative route around the detected failure (Figure 2b, mark 4, bold green arrows). Router S continues to perform this process of encapsulation and flooding of the protected unicast flow until the process of network convergence is completed.

However, before the network completes the convergence and the new shortest paths are calculated, R routers may receive a multicast packets of M-REP flow even through interfaces do not match their current selection of the correct Reverse Path Forwarding (RPF) interfaces. This statement is conditioned by the fact that the current routing tables have not yet been updated with the new information resulting from a link failure.

Incorrect selection of the RPF interface would prevent routers, with the M-REP mechanism implemented, to accept and forward the multicast M-REP flow. For this reason, we created the following modification to the RPF interface selection:

For a router that is not suitable for the M-REP multicast (S, G) flow, let an RPF interface be the one that allows the first multicast packet of a suitable multicast (S, G) flow.

The original PIM-DM communication processing and sending mechanism, as well as legacy PIM-DM RPF selection, are not modified. The revised RPF check will only apply to the specific range of multicast group addresses reserved for the M-REP mechanism.

The term ”first packet“ used in the modified RPF rule (the rule of first-arrival) refers to a multicast packet, the processing of which leads to the creation of a new multicast table entry for a specific (S, G) pair. This principle is in accordance with the standard rule of the PIM-DM protocol that the first multicast packet of a specific (S, G) pair requires the creation of a new entry in the multicast routing table. This record does not exist before the arrival of the first multicast packet of a specific multicast (S, G) pair.

After selecting a RPF interface, previous routers forward multicast packets from their other interfaces and active PIM-DMs. Each router has exactly one RPF interface for a specific multicast M-REP (S, G) pair.

An alternative path created by the M-REP IPFRR mechanism is random because its formation is conditional on the arrival of M-REP multicast packets to individual routers. Consequently, the alternative path created by the M-REP mechanism is not the shortest possible path (Figure 3). However, other IPFRR mechanisms do not generally provide the shortest alternative paths [30,62].

The destination D router may be a provider edge (PE) router or a router that is directly connected to the destination network of the original protected unicast data flow. When multicast M-REP packets arrive at the destination D router, these encapsulated protected flow packets must be restored (decapsulated) to their original format and then routed according to the unicast routing table.

The decapsulation process of the M-REP multicast flow is performed by the destination router D. This router should be directly connected to the destination network of the original unicast packets. To restore the flow back to its original unicast form correctly, destination router D must have the original unicast flow information (source and destination addresses). The M-REP mechanism currently uses a tunneling technique, which means encapsulation of IPv4 unicast communication in new multicast packets (Figure 4). This technique is one of many possible solutions of how we can preserve the original source and destination address of the packet. Another option is to use extension headers (for example in IPv6).

RPF in PIM-DM is a mechanism to ensure that packets are received on a correct interface and to prevent the creation of the micro-loops. For the needs of the M-REP mechanism, the original RPF was modified following the rule of First-Arrival. The research question we faced was whether this modification could cause micro-loops. The verification was performed using the mathematical proof by a contradiction [56], which confirmed that the modification does not cause routing loops. However, the M-REP mechanism operates in topologies which must meet two conditions, the network topology must use point-to-point links only and the target of original protected flow must be directly connected to router D.

The M-REP mechanism will result in exactly one path created between routers S and D. Switching from the alternate M-REP path back to the original unicast path after network convergence is controlled by a dedicated timer. This timer is set to a value greater than the unicast routing protocol convergence time.

It is important to point out that the legacy multicast tree creation procedure used in the PIM-DM protocol remains unmodified. A router with an empty list of output interfaces (OIL) for a specific flow (S, G) logs out of the multicast distribution tree by sending a Prune message from its RPF interface. The Prune message will also be sent out on non-RPF interfaces that received packets fom (S, G) pair.

### 4.2. M-REP State Machine

For a logical representation of the M-REP mechanism, state diagrams are created for S router (Figure 5), as well as D and R routers (Figure 6).

The M-REP process of the mentioned routers can move between the states described in the Table 3:

## 5. Enhancements for M-REP

Although the M-REP mechanism represents a new approach to addressing the IPFRR, this mechanism contains some limitations, which we removed with the proposed extensions. In this section we present the results of further research on M-REP mechanism enhancements. We have primarily focused on the treatment of multiple failures and the enhancement of specific deployments, i.e., the Area Border Router extension.

### 5.1. Multiple Failures

So far, we have dealt with the failure of a single link or router at a time. In critical situations, multiple failures may occur at the same time (Figure 7). This situation is an issue for the M-REP mechanism because it is not able to find an alternative route, although an alternative path exists. Nevertheless, it should be noted, that most of FRR mechanisms analyzed are not able to solve this situation of multiple failures at the given time. Their principles simply do not allow for the correction of multiple failures.

To solve this problem, we propose an extension of the M-REP mechanism, which is called the Swap method. To explain its principle better, we can divide the method into three separate steps. We were inspired by the Multiprotocol Label Switching (MPLS) technology and its functions: Push, Swap and Pop. The step definitions are as follows:Push is defined as the encapsulation of an M-REP packet;Swap is defined as the replacement of a multicast M-REP address by another one;Pop is defined as the decapsulation of M-REP packets, i.e., the decapsulation of original unicast flow from the multicast M-REP packets.

The principle of the EM-REP mechanism in the event of a single failure is unchanged compared to the original design. If another failure occurs at a different time to the original failure, the method Swap shall be used.

This interpretation implies that the used M-REP multicast address (after the first failure at time t) will be in the event of a further failure (at time t + x, where x is the time difference between the first and second failure) replaced by another predefined multicast address. However, this behavior is not efficient and can be optimized further. This implies the following behavior.

A router that detects a new connection failure on already used M-REP backup path of a particular multicast flow will replace the multicast destination address of the existing M-REP header with another multicast address. That is, a router detecting a new failure on the original M-REP backup path becomes the next local repair point, the router S. This will force the router to start a new flooding process but using a different multicast address for the M-REP flow.

This behavior is shown in Figure 8. The primary path for delivering unicast packets from Source to Destination is through R1 → R2 → R5 → D (red arrows). Router R2, which has detected at time *t* the first link failure on the primary path, becomes the router S (Figure 8a, mark 1). Next, router S (R2) begins to encapsulate the protected unicast flow to a specific M-REP multicast flow identified by (S, G) pair, and initiates the flooding process in the topology (Figure 8a, mark 2). Routers that use PIM-DM with modified RPF control, receive multicast traffic (the first-arrival rule) and create an alternative M-REP pathway. The path goes through R2 (S) → R4 → D. Routers on interfaces that do not have receivers for the M-REP multicast flow or receive multicast traffic as the second ones are pruned using Prune messages.

At time t + x, router R4 has detected a second network failure (Figure 8b, mark 3), which occurred on the link through which the alternative M-REP route (S, G) leads. R4 becomes the next source router S2. R4 router replaces the destination multicast address of the original (S, G) multicast flow with the new destination multicast address (S, G + 1). Next, the flooding process in the network starts again (Figure 8b, mark 4). As a result, the new alternative M-REP path is created for the multicast flow (S, G + 1). The path goes through R4 (S2) → R5 → D (Figure 8b, mark 5). The whole resulting alternative M-REP path will go through R2 (S) → R4 (S2) → R5 → D. The part between R2 (S) → R4 (S2) was constructed as the multicast distribution tree for the (S, G) flow. The second part between R4 (S2) → R5 →D was constructed for the multicast flow (S, G + 1).

In the proposed solution, it is necessary to deal only with failures that have already occurred on the alternative route created by the EM-REP mechanism. Failures that occur outside of the alternative path do not affect or interfere with the path created and should not be addressed.

### 5.2. ABR Extension

The PIM-DM protocol, which is used by the M-REP mechanism, assumes that all connected end stations are interested in receiving the multicast traffic. Therefore, the PIM-DM router delivers multicast packets simply by flooding the packets to all active PIM-DM neighboring routers. PIM-DM routers that do not have receivers for a given multicast or have received multicast packets on interfaces, which do not pass RPF control, will be pruned from the distribution tree.

These processes take place in the beginning of multicast broadcasting and periodically later on, so they cause an unnecessary network load [68]. As the EM-REP mechanism uses these PIM-DM processes (flood and prune), they are only carried out until the network convergence process is complete. Subsequently, the routing protocol then takes control of the router’s routing logic. From this point of view, networks consisting of several administrative areas appear to be problematic. In this case, the multicast (S, G) flow flooded by router S will be delivered to all routers in all administrative areas (Figure 9).

Here, we propose the modification of the M-REP behavior applied on border routers of administrative areas (ABR). If we consider a network with applied OSPF routing, the area boundary routers are called the Area Border Router or the Autonomous System Boundary Router (ASBR). In this case, if a failure occurs in a given area, the ABR/ASBR router will act as a decapsulating router instead of the original D router. It means, that the ABR/ASBR router will decapsulate a specific M-REP multicast (S, G) flow back to the original unicast communication. The ABR behavioral design in OSPF is shown in the diagram (Figure 10).

Let us explain this process using the topology shown in Figure 11. The source sends its packets to the destination. R01 detects a link failure to the next-hop router and begins to encapsulate the unicast flow on the M-REP specific multicast flow (S, G). In this case, the boundary routers (ABR/ASBR) are R02, R12 and R21. Using the modified behavior for the ABR/ASBR routers (Figure 10), the specific multicast M-REP flow will not pass to the other areas.

This principle also removes the original M-REP mechanism design requirement, which assumes that the router D is directly connected to the destination. This solution requires that two different areas are connected over one link and only one failure has occurred there.

Routers R12 and R21 will behave according to Action no. 1 (Figure 10). This means that the M-REP multicast will not be forwarded to the next area. The R02 router, however, will behave according to the Action no. 2 (Figure 10), which causes the decapsulation of the multicast M-REP flow and its further delivery to the destination.

### 5.3. Manual Configuration of Router D

Another way to select a router that performs the decapsulation of the M-REP multicast flow back to unicast is to manually configure a router as the decapsulating router (router D). In practice, the network administrator would manually select and configure a router to perform the decapsulation process (Figure 12).

An example of the situation, in which router D has to be manually configured when the destination of the protected unicast flow is not in the domain where a failure occurred is presented in Figure 12 (Manual configuration of Router D). In this case, the administrator must manually configure the Provider Edge Output (PeO) for router function D.

## 6. Evaluation of the EM-REP Proposal

The functionalities of the enhanced version of the M-REP algorithm (EM-REP) proposed in Section 5.1 and Section 5.2 have been verified by simulations. The implementation of the algorithm itself together and its extensions, as well as the creation of testing scenarios, were performed in the OMNeT++ discrete event simulator. The implementation is based on modification of the Automated Network Simulation and Analysis (ANSA) [69] and INET Framework Objective Modular Network Testbed in C++ (OMNeT++) libraries [70]. The ANSA library implements the multicast technology and the INET provides OSPF routing functionalities.

The correct behavior of the enhanced M-REP algorithm functions proposed in the paper has been successfully tested using several scenarios. The scenarios simulate various types of failures for various topologies. In these scenarios, we focus on the correctness of the partial activities of the algorithm, as well as on the investigation of the correct delivery of packets belonging to the protected flow to its destination. In this section, we introduce one of the comprehensive testing scenarios. The topology used in the scenario is shown in Figure 13. As a unicast routing protocol, the OSPFv2 protocol in a multiarea deployment model has been used. The routing domain consists of five OSPF areas, 23 routers and three hosts. For testing purposes, we generate data flow originated from host H11 to the H42 receiver. This data represents a protected stream of user datagrams, the delivery of which is ensured by our algorithm. In the case of a stable and error-free network condition, the delivery of packets is following the shortest path selected by OSPF (in Figure 13 represented by red arrows). In this scenario, we simulate the occurrence of several (three) independent network failures (the improvement introduced in Section 5.1), that occurs inside of different OSPF areas. The purpose of the simulation is to observe how the algorithm will protect user data in the event of multiple network failures within three separated areas.

The description of scenario is as follows. At the beginning of the simulation we wait 200 ms to complete the process of network convergence (i.e., the convergence of OSPF unicast routing), then at the time of 200 sims (simulation seconds) the H11 host starts generating data flow. The source is the H11 with IPv4 address 192.168.11.2, the destination is the host H42 with IPv4 address 192.168.66.2. At the time of 210 sims we simulate the first failure, and the R14 router is shut down. At 212 sims we simulate the second failure as a permanent connection failure between routers R13 and R16. Finally, at 215 sims we simulate another failure of router R05 (Table 4).

### 6.1. Simulation Process: Algorithm Behavior

After the simulation has been started, at 200 sims the H11 host starts to generate packets of protected flow with destination address of H42. This flow, as we have already mentioned, is called a protected flow because the routers in M-REP are configured to encapsulate and flood its packets around the point of failure on the way to the destination. Therefore, at the time of 210 sims, we simulate the first failure inside area 1, where we turn off the router R14. As the R14 router is on the best path to the destination, the OSPF will begin to flood its OSPF Link-State Advertisement (LSA) updates and will start to converge. However, until the convergence ends, the R12 router quickly detects that its neighbor has failed (they have either direct connection or use the BFD mechanism), and R12 becomes the source router (S router). That is, R12 begins encapsulating the unicast packets of the protected flow into new multicast packets of the (S, G1) pair, since it is configured to protect the flow from 192.168.11.2 to 192.168.66.2. When creating a multicast packet header, the router will use the original IP address as the source address S, i.e., the IP address of sender (H11). As the multicast destination address G1, the router will use the predefined M-REP multicast address (unique and configured for each protected flow); here it is 226.1.1.1. This behavior is shown in Figure 14. The original unicast destination address is stored for future decapsulation in a variable named MREPdestAddress. The multicast packet is then immediately flooded out using the PIM-DM mechanism. As a result, due to the M-REP algorithm modification (the rule of first arrival), the PIM-DM will construct an alternative path around the failure. The new path lead through R11 → R12 → R13 → R16 in this area.

At 212 sims, we have scheduled the second failure, in which the connection between R13/R16 will be interrupted. The link is already a part of the alternative route (distribution tree) constructed before. Here we have both, a second error and a protected flow, which is once encapsulated. The moment this failure occurs, R13 becomes the new source router S2. R13 as S2 detects that it is already on the previously constructed repair path of the first M-REP run (according to its record that includes interfaces list for (S, G1) tree). Therefore, for each multicast packets of (S, G1) pair, the router R13 replaces the previous M-REP G1 multicast address (226.1.1.1) by the new one, G2 (226.1.1.2, Figure 15). This event triggers a new flooding process, creating a new alternative path around the second failure. In this case, the path leads through the routers R11 → R13 → R15 of the area 1.

Our enhancement of the M-REP algorithm assumes, that the decapsulation of original unicast packets of a protected flow from the carrier multicast packets is performed on an ABR router. The ABR selection is performed according to the conditions specified in Section 5.2, which in our case is the R16 router. The modification proposed in Section 5.2 stops the flooding of M-REP packets from one area to other areas, except area 1. Router R16 replaces the destination address of the G2 M-REP packets back to the original unicast one (i.e., 192.168.66.2 taken from MREPdestAddress header, Figure 16). This converts the multicast communication back, and the flow from H11 to H42 is routed as unicast again.

The third and final simulated failure is scheduled at the time of 215 sims. The failure represents an error of the R05 router, that is located inside of the backbone area 0. Here, the process of encapsulating and PIM flooding is repeated. When this failure occurs, R01 becomes the next source router S and begins encapsulating packets of the protected flow using the predefined M-REP multicast address. The multicast address, as we mentioned, must be configured for the protected flow. The failure occurred in a different OSPF area than in the previous two cases. Our proposed solution reduces the flooding from one OSPF area to another, so that M-REP can use the same multicast destination address as in area 1, i.e., the multicast address 226.1.1.1.

Table 5 displays the output from OMNeT++ simulation, which shows how packets are handled in the moment of the R05 router failure, and which destination addresses are used to deliver packets of the protected flow. The M-REP constructs an alternative route that leads through routers R01 → R03 → R04.

### 6.2. Evaluation of the EM-REP Mechanism

The main advantage of the M-REP IPFRR mechanism is that the algorithm does not depend on precomputations and even on the unicast routing protocol used. Respecting these properties, we may argue that the M-REP IPFRR mechanism is unique compared to the analyzed ones, as well as other existing IPFRR solutions. The extensions of the M-REP proposed in this paper solve several limitations that have not been resolved in legacy M-REP proposal.

The enhancement described in Section 5.1 has introduced recovery mechanisms that support fast reroute in the event of multiple and persistent connection and node failures thorough the whole network. In situations when subsequent and recurring errors occur concurrently over time, the EM-REP mechanism encapsulates protected flow into several specific multicast distribution trees, or (S, G) traffic groups. For each multicast distribution group, its router S as the root of the distribution tree encapsulates and floods packets through any of its functional links to all PIM-DM neighbors. Therefore, even if a link or node failure occurs in several places, if there is still at least one possible path from the source S to the destination D, the EM-REP mechanism can find it and use it as an alternative path. This is a unique behavior of EM-REP that provides the 100% repair coverage. This behavior was among other ones simulated by the complex scenario just described. Here we have simulated three consecutive network failures. The simulation results confirmed expected core EM-REP behavior (detection, encapsulation, and flooding), as well as the extended protection against multiple failures, as we proposed in Section 5.1. The EM-REP mechanism has constructed several distribution trees and protects the data of the specified protected flow from multiple failures. The mechanism ensures that all packets of the protected flow were delivered to its destination D.

The second enhancement, which was described in Section 5.2, addresses the issue that in the original M- REP mechanism, the destination host has to be directly connected to a network with a router that performs decapsulation (router D). The extension also reduces a flooding process of M-REP packets in a network with multiple routing areas.

Compared to other existing IPFRR mechanisms, the concept of the M-REP mechanism (as well as its enhanced version EM-REP) brings several advantages in addition to the mentioned ones. Some of its drawbacks are also known. An overview of advantages and disadvantages of the EM-REP mechanism is given in the Table 6.

A more accurate comparison of the selected features with other existing IPFRR mechanisms is provided in Table 7 below. As we can see, the EM-REP mechanism is unique in several specific areas.

### 6.3. Time of Repair: Algorithm Speed

The network recovery time usually consists of two parts. The first part consists of the time in which a router is able to detect the failure of its link, or the unavailability of its connected neighbors. In practice, a specialized protocol is usually used for this purpose. The most used protocol is the Bidirectional Forwarding Detection (BFD), a protocol standardized by Internet Engineering Task Force (IETF) in Request for Comments (RFC) 5880. Using BFD, the router can detect a connection failure with a neighboring node in less than 30 ms, depending on the timer settings. Specifically, the mentioned time of 30 ms can be achieved by setting the hello interval to 10 ms. If no hello message is received from a neighboring node within three hello intervals, the BFD session with that neighbor is declared invalid, i.e., the neighbor is considered unavailable. It is the state of unavailability that subsequently triggers an IPFRR mechanism.

The second part, which defines the recovery time, is the amount of time required to create an alternative path and resume an interrupted communication. This time depends mainly on the speed of the specific IPFRR mechanism (or its algorithm). Current FRR mechanisms operate using a proactive approach. This means that all alternative paths for all possible destination are calculated in advance before the outage itself occurs. These preliminary calculations differ in their computational complexity and time depending on the IPFRR mechanism used. At the same time, different IPFRR mechanisms have different requirements regarding the required space needed to store their results. However, once a failure is detected, the installation and use of a pre-calculated alternative path is immediate. Compared to link failure detection, this time is minimal and negligible.

However, the EM-REP mechanism operates in principle in a reactive manner. An alternative path is created randomly as the result of flooding and pruning mechanisms used by our modified PIM-DM (i.e., EM-REP). The distribution path (single-branched tree), as has been already mentioned, is constructed using the principle of the first arrival of packets, i.e., packets that arrive first on individual router interfaces after the flooding process. This subsequently creates interfaces of the first arrival (PIM-DM RPF ports) and a chain of routers of an alternative path. Simulations show the speed of network recovery achieved by the EM-REP mechanism is comparable, respectively the same, as the network recovery speed achieved through the proactive FRR mechanisms. The main difference is that the EM-REP mechanism does not require preparatory calculations or additional router resources.

On the other hand, the network load is initially higher, as is the case with the proactive FRR mechanisms. This is the result of the initial EM-MREP flooding process. However, the EM-REP was not designed to protect all flows affected by a failure. EM-REP protects only specific but important customer data flows that require special treatment or lossless delivery through an ISP. These flows are only a subset of all flows affected by the error. In addition, we expect that the EM-REP mechanism, like other IPFRRs, will only work for a short time, not longer than a few tens of milliseconds or a few seconds. It ends when the network convergence process is complete, the multicast tree is no longer used, and packets of protected flows are routed again as unicast packets.

## 7. Conclusions

The paper presents the Enhanced Multicast Repair (EM-REP) FRR mechanism, which solves several limitations of the legacy M-REP FRR mechanism. This means mainly support for fast reroute in the event of continuous link and node failures throughout the whole network and that the destination host does not have to be directly connected to a network with a router that performs decapsulation, which also reduces a flooding process of M-REP packets in a network with multiple routing areas.

Both mechanisms belong to the family of Fast Reroute solutions. The EM-REP mechanism presented in this paper, makes it possible to create an alternate backup path that allows packets to bypass the failures of one or more links or nodes at a given time. To achieve this goal, the EM-REP has been built on two cornerstones, the PIM-DM protocol and tunneling. The PIM-DM delivers multicast data thorough a distribution tree constructed by the flooding and reverse pruning. The EM-REP is based on this behavior, where in the event of a failure (or even multiple failures), for specific traffic flow an alternate path is built by PIM-DM flooding and pruning. In this case, the router begins encapsulating unicast packets of the protected flow into multicast packets flooded out and around the failure. To ensure correct operation of routers, we have not modified the PIM-DM process as such, i.e., for a common multicast traffic the PIM-DM process works as usual. However, for the correct construction of alternative FRR paths (distribution trees), we have modified the PIM-DM RPF process, where we use the rule of the first arrival. Alternative paths are created only for protected flows, so the router must identify correct packets in some way. In short, we expect that flow identifiers are predefined and preconfigured by the network administrator in advance. However, in future, some dynamic distribution mechanism may be used, inspired, for example, by those used for a dynamic distribution of Rendezvous Point (RP) addresses (Auto-RP, BSR Bootstrap Router mechanism), but here used for the distribution of protected flow identifiers.

As has been already mentioned, most FRR mechanisms require the pre-calculation of alternative routes for different network failure scenarios. On the one hand, these preparatory calculations have undesirable effects on the router’s limited resources, such as CPU load and memory. On the other hand, they may depend on a specific link-state routing protocol. The EM-REP mechanism does not require any preparatory calculations, which is effective for IoT devices such as sensors.

Moreover, the EM-REP does not depend on any unicast routing protocol. In addition, although EM-REP can bring benefits resulting from the use of a specific routing protocol supporting the organization of unicast routing to areas, as is presented here for OSPF, it could work for IS-IS as well. Furthermore, the EM-REP FRR mechanism provides 100% repair coverage for single as well as multiple failures occurring at different times and places in the network. Finally, EM-REP eliminates the condition of directly connected destination to router D of legacy M-REP.

The EM-REP mechanism uses the generic flooding process of the PIM-DM protocol to provide the protection for specific flows that expect special handling inside the network. The behavior and goal are generic enough with a wider application domain. In the area of WSN and the IoT, it can be used to distribute, for example, urgent messages across the WSN network or to assure the time-critical delivery of important information from sensors to gateways or behind to analytic servers.

The EM-REP was fully implemented, and its correctness was tested using the OMNeT++ simulator. We have performed extensive tests of the implementation in different networking scenarios, which validated the functional correctness of all the mechanism functions. The principle of the mechanism is unique, and it is possible to apply it in other networks such as WSN, IoT architecture, and other areas as well, which will be studied in future work.

## Figures and Tables

**Figure 1 sensors-20-03428-f001:**
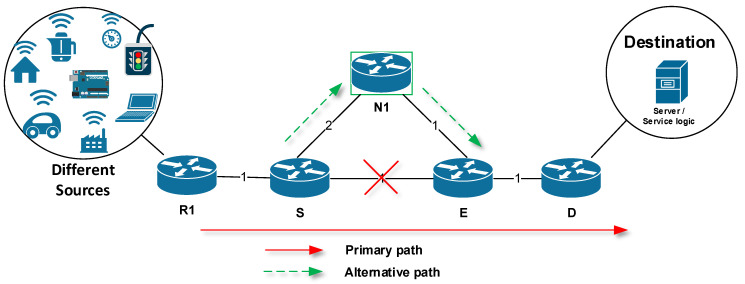
The principle of Fast ReRoute (FRR).

**Figure 2 sensors-20-03428-f002:**
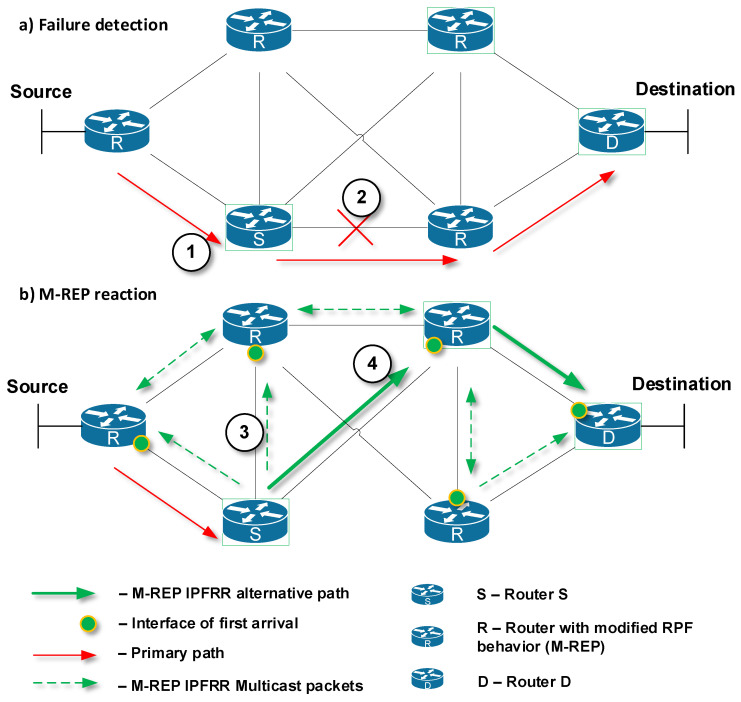
Principle of the M-REP mechanism

**Figure 3 sensors-20-03428-f003:**
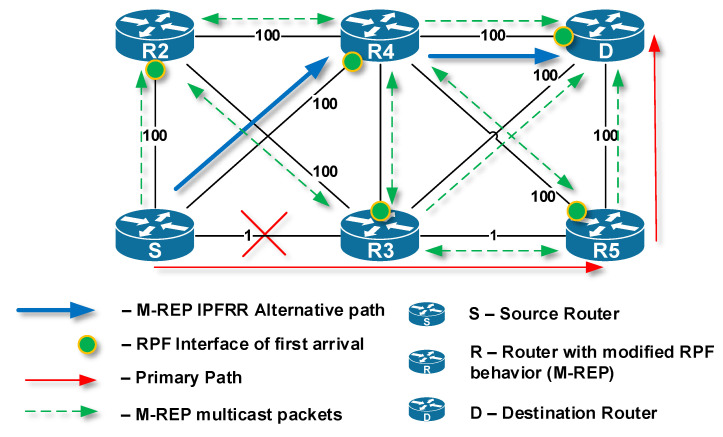
The first arrival rule.

**Figure 4 sensors-20-03428-f004:**
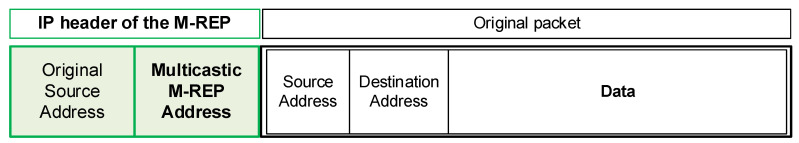
The M-REP encapsulation of the unicast communication.

**Figure 5 sensors-20-03428-f005:**
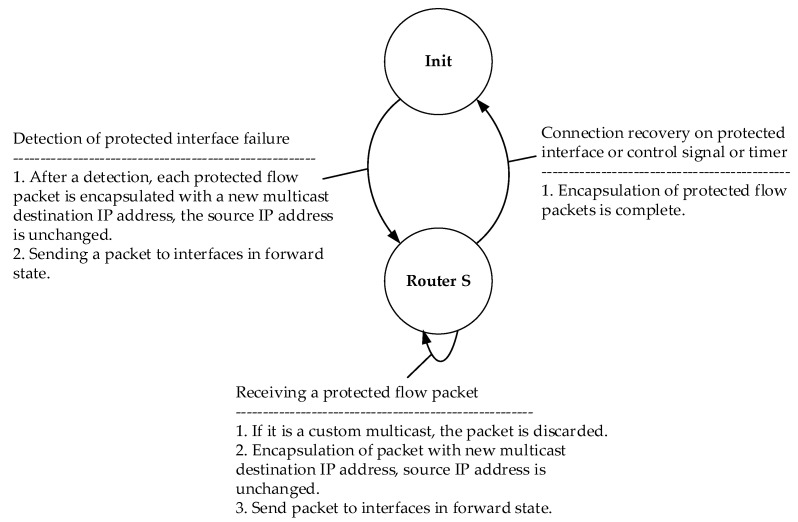
The M-REP mechanism state diagram of router S.

**Figure 6 sensors-20-03428-f006:**
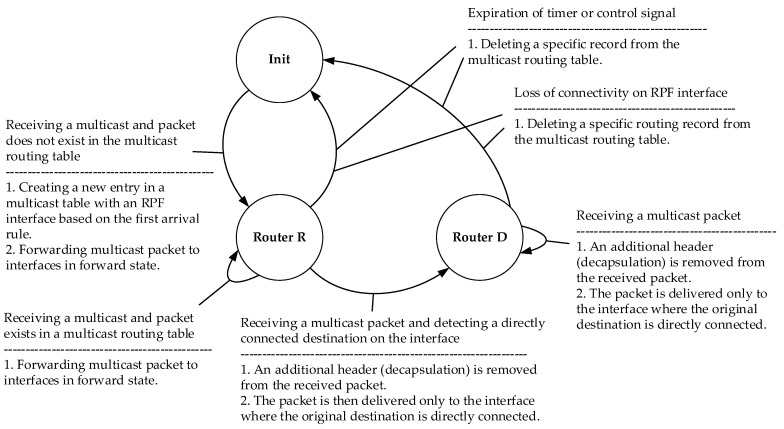
The M-REP state diagram for D and R routers.

**Figure 7 sensors-20-03428-f007:**
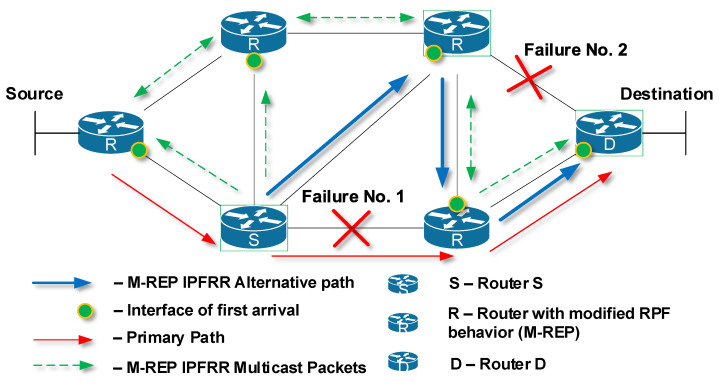
M-REP: reroute in the event of multiple failures.

**Figure 8 sensors-20-03428-f008:**
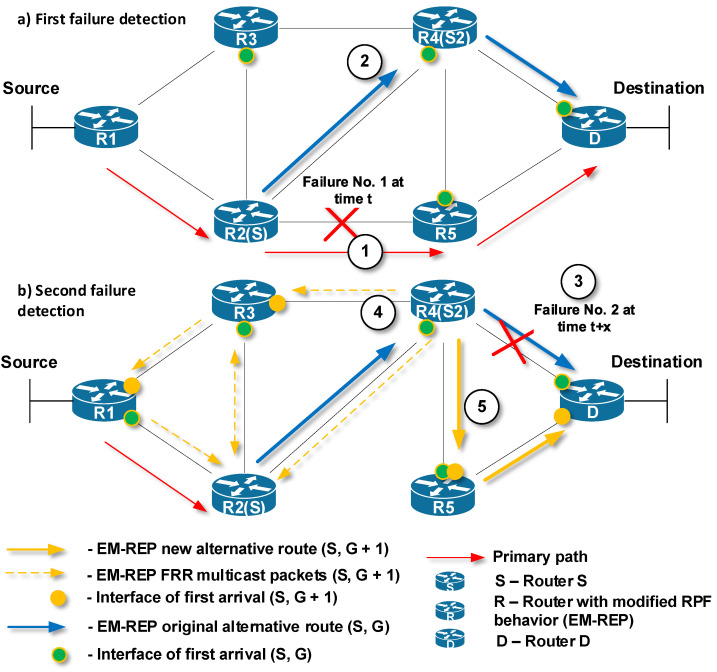
The EM-REP principle at multiple failures at different times.

**Figure 9 sensors-20-03428-f009:**
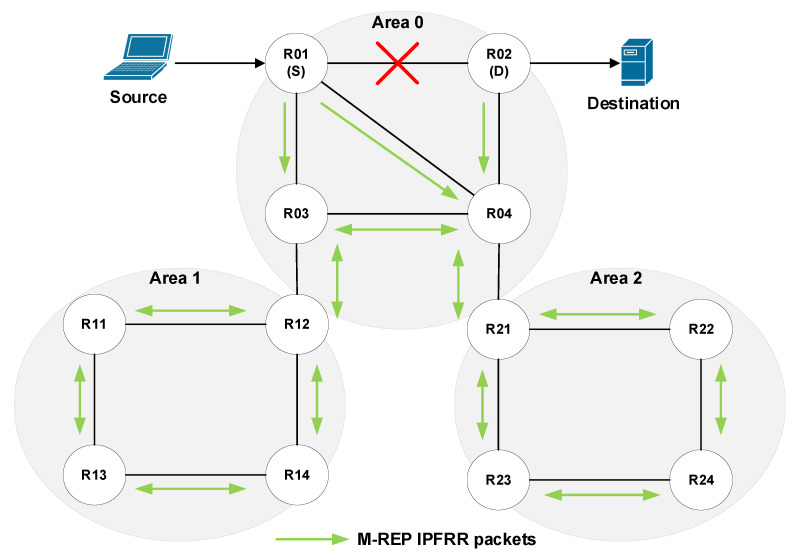
The process of PIM-DM flooding in multi-area Open Shortest Path First (OSPF).

**Figure 10 sensors-20-03428-f010:**
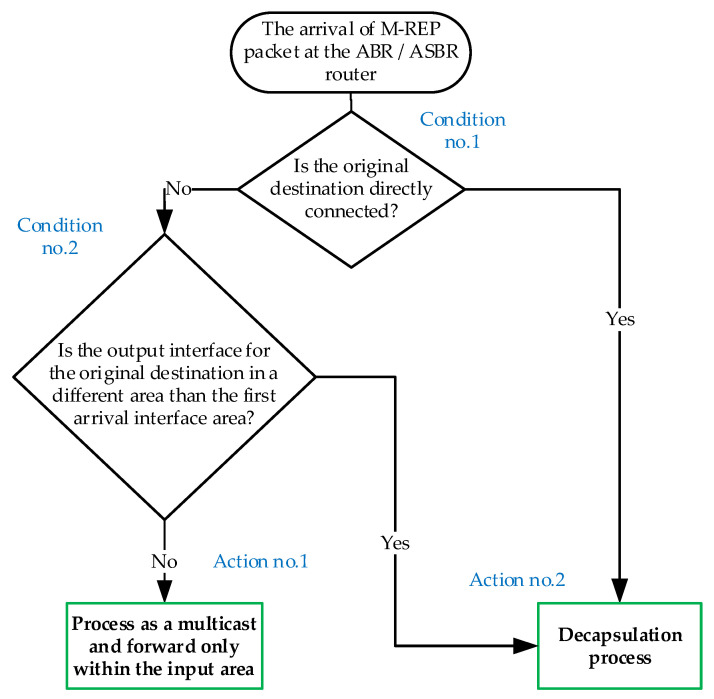
The Area Border Router (ABR) router behavior after the arrival of M-REP packet.

**Figure 11 sensors-20-03428-f011:**
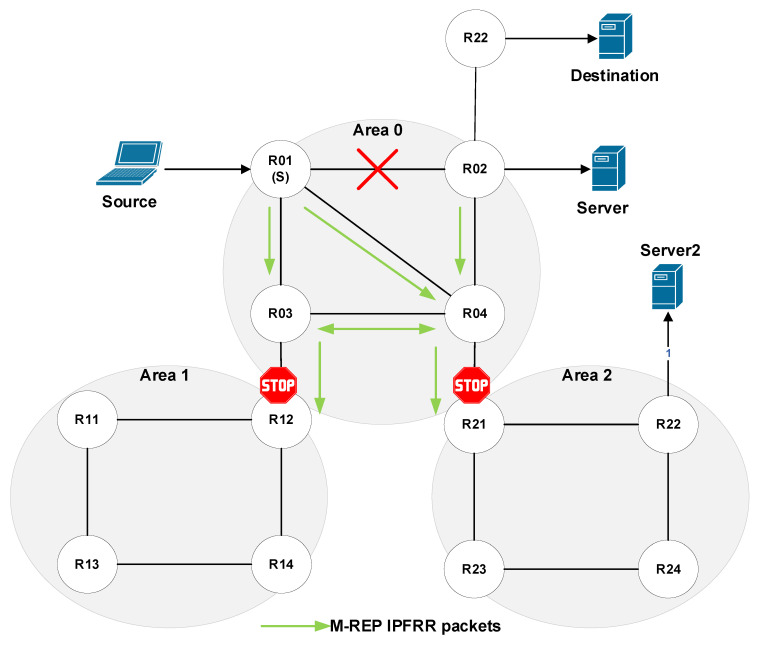
The flooding process with modified ABR.

**Figure 12 sensors-20-03428-f012:**
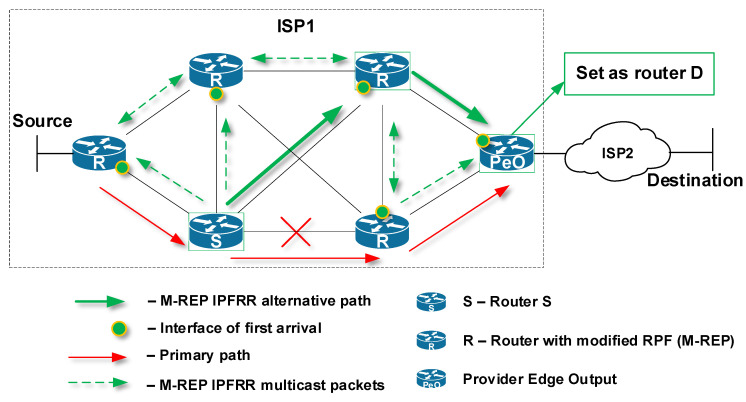
Manual configuration of Router D.

**Figure 13 sensors-20-03428-f013:**
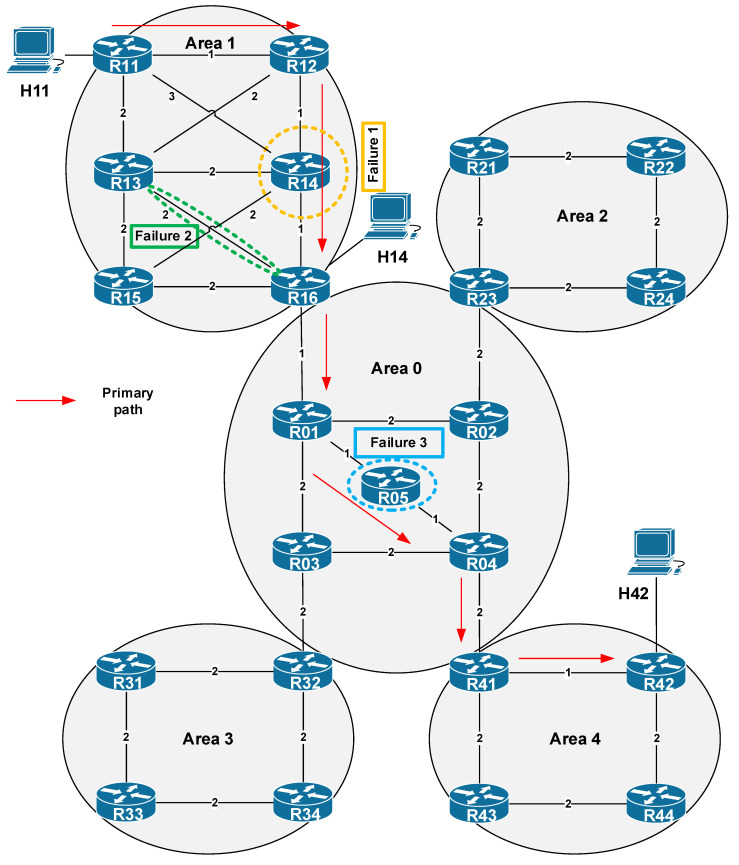
The OMNeT++ simulation topology.

**Figure 14 sensors-20-03428-f014:**
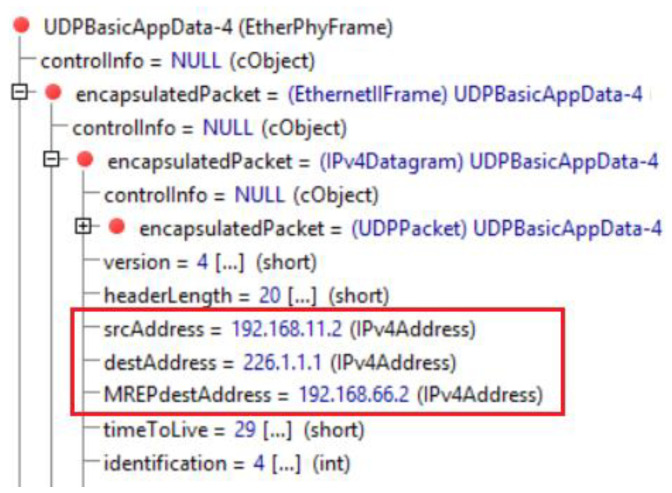
Encapsulated EM-REP packet.

**Figure 15 sensors-20-03428-f015:**
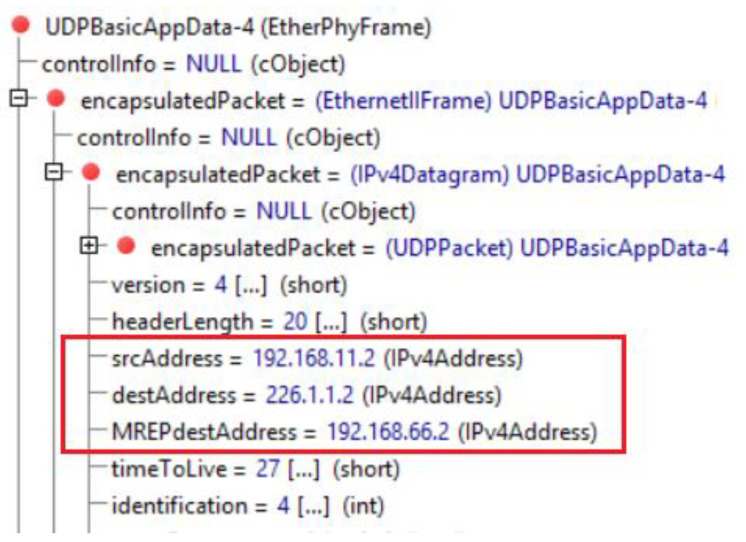
Second EM-REP run.

**Figure 16 sensors-20-03428-f016:**
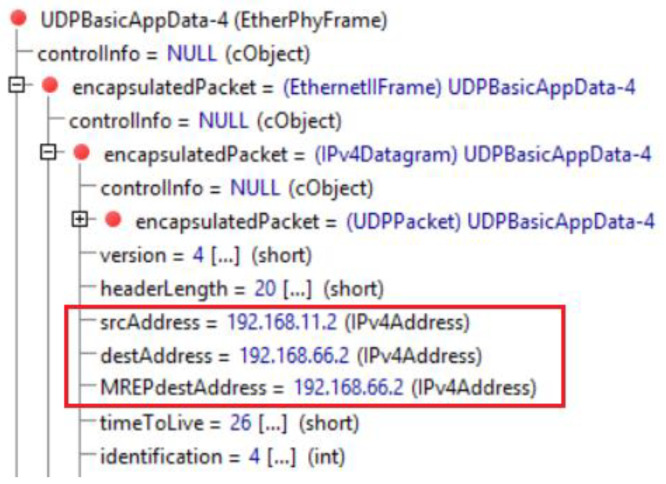
Restoration of EM-REP packet to original destination address.

**Table 1 sensors-20-03428-t001:** FRR mechanisms: features comparison.

FRR Mechanism	100% Repair Coverage	Precomputing	Packet Modification	Dependency on Link-State Routing Protocols
ECMP FRR	No	Yes	No	No
BIER-TE (M)	Yes	Yes	Yes	No
Directed LFA	Yes	Yes	Yes	Yes
LFA	No	Yes	No	No
MoFRR	No	Yes	No	No
MPLS-TE FRR	No	Yes	Yes	No
MRC	Yes	Yes	Yes	Yes
MRT	Yes	Yes	Yes	Yes
Not-Via Addresses	Yes	Yes	Yes	Yes
Remote LFA	No	Yes	Yes	Yes
TI-LFA	Yes	Yes	Yes	Yes

Notes: ECMP—Equal-Cost Multi-Path Routing; BIER-TE—Bit Index Explicit Replication—Traffic Engineering; LFA—Loop-Free Alternate; MoFRR—Multicast-Only Fast Reroute; MRC—Multiple Routing Configurations; MRT—Maximally Redundant Trees; TI-LFA—Topology Independent Loop-Free Alternate.

**Table 2 sensors-20-03428-t002:** Terminology used by the Multicast Repair (M-REP) mechanism.

**Protected Flow**	Unicast flow of packets with specified source and destination IP addresses (source, destination). The M-REP protects packets of a secured flow from losses during network failures. The unicast source address specifies a sending host, the unicast destination address specifies the receiving host.
**M-REP Address**	Special reserved multicast address used exclusively by the M-REP mechanism. The address represents a multicast G address of PIM-DM (S, G) pair, that is reserved and preconfigured for each protected flow. Each protected flow has unique G multicast address.
**M-REP Flow**	A multicast (S, G) flow that encapsulates packets of a protected flow in the event of a failure. The S address is the original source IP address of the protected flow. The G address is the multicast M-REP address. Together, they define a multicast distribution (Source, M-REP address) pair.
**Received Packet**	A packet of a protected flow received by a router. The router identifies the packet based on configured IP addresses of a protected flow.
**Received Multicast Packet**	A packet of the M-REP flow received by a router. The packet is identified by its M-REP address destination address.
**Protected Interface**	Router output interface selected according to the unicast routing table used. The interface is used for the routing of a protected flow (destination).
**Failure of Protected Interface**	Loss of connectivity on the protected interface.
**Reverse-path forwarding (RPF) Interface**	A router interface that first receives a multicast packet with the specified destination M-REP address (M-REP address). This interface has a similar role to the RPF interface in the “original” PIM-DM specification. Each router may have at most one RPF interface per M-REP address.
**Connected Destination**	The network that contains the host with a protected flow destination address. The D router is directly connected to this network by one of its interfaces.
**M-REP Requirements**	Point-to-point routers. The original destination of the original unicast communication must be directly connected to router D.

**Table 3 sensors-20-03428-t003:** States of the M-REP mechanism.

**State:**	**Any Condition**
Event:	-
New state:	Init
Action:	Initializing the M-REP mechanism on the router. The mechanism is initialized only for the first time. After this action, it is in the monitoring mode.
**State:**	**Init**
Event:	Failure of a protected interface
New state:	Router S
Action:	If the router detects a connectivity failure on the output interface during the processing of protected packet flow (defined by source, destination addresses), it becomes router S. After the failure is detected, all packets within a protected flow are encapsulated with an additional packet header (source, M-REP add). Router S does not have an input RPF interface for the multicast flow, which means that it discards the packet (s) with the destination multicast address (M-REP add). Note: Deactivation of the M-REP mechanism can also be performed using a timer set to a time, which will ensure that the convergence process in the network has completed. In this case, the timer starts when the encapsulation starts.
**State:**	**Router S**
Event:	Recovery of connection on protected interface or control signal or timer.
New state:	Init
Action:	The router stops encapsulating the protected flow and enters the Init state.
**State:**	**Init**
Event:	Receiving a multicast packet and no entry in the multicast routing table.
New state:	Router R
Action:	The router has received a packet with the multicast address (M-REP add) and does not have a directly connected destination. If the router does not have an entry in its multicast routing table for (source, M-REP add) pair, it creates a new entry with the RPF interface that has first received the multicast packet. The RPF interface is just one. Interfaces other than RPF and with active PIM-DM, become output interfaces. Received multicast packet is then forwarded to all output interfaces. If the router has a multicast routing table entry for (source, M-REP add) pair and has received a multicast packet on the RPF interface, the packet is forwarded to all PIM-DM output interfaces. If the router has a multicast routing table entry for (source, M-REP add) pair and has received a multicast packet on the NON-RPF interface, the multicast packet is dropped.
**State:**	**Router R**
Event:	Receiving multicast packet and destination is directly connected on an interface.
New state:	Router D
Action:	The router has received a multicast packet (the multicast address M-REP add is used) and has directly connected destination. Router D is a router that has the original destination directly connected to one of its interfaces. Multicast header is then removed from the received multicast packet, which means that packet is decapsulated and returned to its original state. After decapsulation, the packet is sent out through the interface where the directly connected destination is located. Interfaces other than the RPF interface will send a Prune message.
**State:**	**Router R, Router D**
Event:	Timer expires or control signal
New state:	Init
Action:	Deletes the entry in the multicast routing table. After this action, mechanism moves to the Init state.
**State:**	**Router R, Router D**
Event:	Loss of connectivity on an RPF interface.
New state:	Init
Action:	Deletes the entry in the multicast routing table and move to the Init state.

**Table 4 sensors-20-03428-t004:** Test description.

Time	Description of Action
<200	Time necessary for the OSPF convergence and stabilization of network processes.
200	H11 starts the flow
210	Router R14 failure
212	Drop of link R13 / R16
215	Router R05 failure

**Table 5 sensors-20-03428-t005:** The result of the third failure simulation.

Time	Source/Destination	Name	Destination Address
215.00007242	→R01	UDPBasicAppData-185	192.168.66.2
215.00008484	R01 → R02	UDPBasicAppData-185	226.1.1.1
215.00008484	R01 → R03	UDPBasicAppData-185	226.1.1.1
215.00009726	R02 → R04	UDPBasicAppData-185	226.1.1.1
215.00009726	R02 → R23	UDPBasicAppData-185	226.1.1.1
215.00009726	R03 → R32	UDPBasicAppData-185	226.1.1.1
215.00010968	R04 → R41	UDPBasicAppData-185	226.1.1.1
215.0001221	R41 → R42	UDPBasicAppData-185	192.168.66.2
215.00013452	R42 → H42	UDPBasicAppData-185	192.168.66.2

**Table 6 sensors-20-03428-t006:** Features of M-REP mechanism.

Advantages	Disadvantages
No pre-computation Suitable for networks of any size Independence of unicast routing protocols in general, but with optimized feature set when using OSPF 100% repair coverage Support of multiple failure repairs at the same time Fix multiple failures at different times (solution presented in Section 5.1) Relatively easy implementation through PIM-DM modification	Does not support multiaccess network segments, i.e., only point-to-point links are supported Random alternative route (hard to predetermine) Packet modification (tunneling) Flooding/pruning process of PIM-DM distribution path

**Table 7 sensors-20-03428-t007:** Comparison of the innovative M-REP mechanism with existing solutions.

Title	100% Repair Coverage	Precalculations (Precomputing)	Packet Modification	Dependency on Link-State Routing Protocols
EM-REP	Yes	No	Yes	No
ECMP FRR	No	Yes	No	No
BIER-TE (M)	Yes	Yes	Yes	No
Directed LFA	Yes	Yes	Yes	Yes
LFA	No	Yes	No	No
MoFRR	No	Yes	No	No
MPLS-TE FRR	No	Yes	Yes	No
MRC	Yes	Yes	Yes	Yes
MRT	Yes	Yes	Yes	Yes
Not-Via Addresses	Yes	Yes	Yes	Yes
Remote LFA	No	Yes	Yes	Yes
TI-LFA	Yes	Yes	Yes	Yes

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
