# Peer review of "Enhanced Multicast Repair Fast Reroute Mechanism for Smart Sensors IoT and Network Infrastructure"

_sensors, 2020, doi:10.3390/s20123428_

Round 1

Reviewer 1 Report

1) Related works analysis should contain both practical and theoretical state-of-art solutions.

2) In turn, the theoretical background of the mechanisms implemented in the EM-REP FRR would significantly strengthen the article.

3) It is suggested to include more detailed research of the repair coverage feature of the presented mechanism.

4) The authors did not provide the study of the impact of the flooding and additionally generated traffic during EM-REP FRR performing in a case of a failure on the network.

5) Using the proposed mechanism in IoT should be explained more specifically.

6) The proposed solution does not take into account the causes of network failures (links, nodes). While it is indicated that IoT networks are most prone to failures due to security attacks.

Reviewer 2 Report

Thanks for the submission.

Overall I would recommend to describe the implementation in more detail, so it becomes more obvious how you selected the evaluation criteria and the reader can better understand the proposed solution at all.

Other comments:

  • Section about related work should clearly state at the end what your solution aims on in comparison to mentioned related work. This should be picked up in the evaluation again.
  • In general if you refer to a figure or table in brackets I would write it the following way (cf. Figure X).
  • Section 4.1 I would formulte in paragraphs and not as a listing. It looks strange somehow.
  • Table 2: Why is the line "protected flow" written in bold compared to the other entries?
  • Lines 308-309 show a citation. But from where does it come? The source is missing
  • Figure 4 rises the question why parts are highlighted in green. Are these parts very important or why?
  • Lines 379-380: Try to avoid that two section titles stand next to each other without text in between.
  • Figure 10 is really nice. Please use the same spelling of teh blue words in text
  • Figures in general are very nice, but complex. Keep in mind that the reader needs to understand them without much effort. On the other hand, sometimes the available text is to short to understand it complete. so improve.
  • Table 4 misses in text the reason why you selected the parameters the way you did
  • Table 7 compares your approach with the related work, but why do you expected it is better than the others? In which way? I can find 4 of the listed related work that have in all columns a yes and your solution only in to. So is it really better?

Reviewer 3 Report

  1. The reviewer mainly concern that this article considers an enhanced multicast repair fast reroute mechanism, but not consider the wireless link failure, which is necessary for IoT.
  2. What if an ABR fails? Assume there is a network consists of three administrative areas, named Area_1, Area_2 and Area_3. They have a public router in pairs, named ABR_12, ABR_13 and ABR_23. Now, the source host is in Area_1 and the destination host is in Area_2. The source sends packets to the destination through ABR_12. Suddenly, ABR_12 fails. According to Section 5.2, ABR_13 will not forward the M-REP multicast to the next area, which leads to the destination unreachable. However, we know that it’s not reasonable.
  3. As we can see, the authors considered only statistic network, how about dynamic network with opportunity routing strategy? It’s more realistic and more challenging.
  4. As an enhanced version of M-REP FRR mechanism, did the authors sacrifice time for space, or space for time? Or the new algorithmn is better in both aspects? The authors should provide experiment data to convince it.
  5. There are many grammar mistakes. In line 183, “the router delivers the packet to the destination over a longer path that is still loop-free an bypasses the network failure”, it seems to be an “and” rather than “an”. In line 390, “we can divide the method functionality of into three separate steps”, the word “of” is superfluous.

Round 2

Reviewer 3 Report

  1. Check the sentence in line 560-561.
  2. Fig.1 needs considering smart sensors IOT.
